# Overview and Recent Advances in Hyphenated Electrochemical Techniques for the Characterization of Electroactive Materials

**DOI:** 10.3390/ma16124226

**Published:** 2023-06-07

**Authors:** José Juan García-Jareño, Jerónimo Agrisuelas, Francisco Vicente

**Affiliations:** Department of Physical-Chemistry, University of Valencia, C/Dr. Moliner 50, 46100 Burjassot, Spain; jeronimo.agrisuelas@uv.es (J.A.); francisco.vicente@uv.es (F.V.)

**Keywords:** hyphenated techniques, crossed derivative functions, crossed transfer functions, EQCM, electrochemical techniques

## Abstract

A hyphenated electrochemical technique consists of the combination of the coupling of an electrochemical technique with a non-electrochemical technique, such as spectroscopical and optical techniques, electrogravimetric techniques, and electromechanical techniques, among others. This review highlights the development of the use of this kind of technique to appreciate the useful information which can be extracted for the characterization of electroactive materials. The use of time derivatives and the acquisition of simultaneous signals from different techniques allow extra information from the crossed derivative functions in the dc-regime to be obtained. This strategy has also been effectively used in the *ac*-regime, reaching valuable information about the kinetics of the electrochemical processes taking place. Among others, molar masses of exchanged species or apparent molar absorptivities at different wavelengths have been estimated, increasing the knowledge of the mechanisms for different electrode processes.

## 1. Introduction

Electrochemical techniques are characterized by providing very precise information on the behavior of certain materials. Most electrochemical techniques work by applying an electrical perturbation to the system under study at a specific applied potential or by forcing an electric current through it, and the response of the system is measured in current or potential. The response to the electrochemical perturbations depends on many factors, such as the kinetics and thermodynamics of the systems under study, but it also depends on the structure and surface area of the system under study, the spatial arrangement of the electrodes, the conductive characteristics of the medium, etc. During electrochemical processes, changes can occur on the electrode’s surface, growth of passive layers, viscoelastic properties, the size of layers, the interface and regions close to the electrode, etc. Sometimes, there is not a single process that provides an electrical signal, but rather we have coupled reactions that may appear as separated or overlapped responses.

The problem lies in the fact that this information always comes to us together with the purely electrical response of the system, which can be interpreted based on a theoretical model or as a deviation from the ideal or expected behavior of the system. An important advantage of electrochemical techniques is that these techniques allow the simultaneous recording of other signals, whose measurement has little or no effect on the electrochemical response. Considering the complexity of the above-mentioned electrochemical processes, this additional information helps to discern different possible mechanisms. Some of these techniques that have been coupled to electrochemical ones are as follows:The electrochemical quartz crystal microbalance (EQCM) devices in the *dc*-regime (electrogravimetry) and in the *ac*-regime (*ac*-electrogravimetry or mass impedance). The EQCM electrode consists of a thin metal layer deposited on a thin sheet of quartz crystal whose resonant frequency is correlated with the mass on the metal electrode layer by Sauerbrey’s equation [1,2,3,4]. This technique has been used to characterize the ionic or solvent insertion in electroactive films [5,6,7,8,9], corrosion processes [10,11,12,13,14], or the growth of passive layers [15,16,17], among others;Spectroscopy (UV-Vis, near IR, Raman). Working in transparent electrochemical cells allows recording the spectra at different wavelengths during an electrochemical experiment. This is known as spectroelectrochemistry and has been widely used to identify changes in the structure of electroactive materials during electrochemical experiments [18,19,20,21,22,23]. Coupling spectroscopy with electrochemical impedance spectroscopy is also possible; this is the color impedance technique [24,25,26,27,28];Digital Video Electrochemistry (DVEC). This technique consists of coupling an electrochemical experiment with the recorded video of the electrode’s surface, allowing the spatiotemporal analysis of the color changes at any part of the electrode’s surface. If compared with spectroelectrochemistry, it has a small resolution but allows an estimation of the homogeneity of the electrode’s surface through the analysis of the color dispersion [29];Mirage effect: changes in density during the ion flux near the electrode’s surface cause a change in the refraction index, and from these changes, the flux of ions can be estimated [30,31,32];Acoustic impedance. The measurement of electromechanical changes is useful for the characterization of the electrode/solution interface and viscoelastic properties of the electrode [33,34].

There are other possible coupled techniques: chromatography; potentiometry; mass spectrometer; etc. We will focus our attention on techniques allowing fast data acquisition simultaneously with the electrochemical experiment. The key point is that the signal from the different coupled techniques is simultaneously measured. Achieving synchronization is crucial for comparing signals, computing time derivatives, and verifying the accuracy of crossed derivatives to determine such parameters as molecular mass or molar absorptivity. This coupling between techniques yields valuable information, increasing the comprehension of the mechanism, kinetics, and processes occurring. The objective of this review is to show recent advances in electrochemical techniques and experimental data analysis methodologies for the characterization of electroactive materials.

## 2. Electrochemical Techniques

During this experiment, a perturbation in potential or current is applied to the electrochemical cell, and the response in current or potential, respectively, is measured and recorded. Various techniques could be used to introduce the perturbations into the system, such as potential or current steps, potential sweep voltammetry, electrochemical impedance spectroscopy, zero current potentiometry, polarization curves, etc. The analysis of the response gives thermodynamics and kinetics information about the studied system. Electrode potentials are directly related to the thermodynamics of the system, while the measured current is a direct measure of the electrochemical reaction rate. Faradaic current can be expressed as the time derivative of electric charge, and the electric charge is directly related to the number of mols reacting on the electrode’s surface. We can express it as the change in the number of mols of the reduced or oxidized forms during a Faradaic electrochemical reaction:(1)$$i=\frac{dq}{dt}=-\sum^{\;}n_{j}F\frac{dn_{Red,j}}{dt}=\sum^{\;}n_{j}F\frac{dn_{Ox,j}}{dt}$$

This is why potential controlled techniques are the preferred ones for the data analysis in hyphenated techniques, as explained below. In this expression, i refers to current, q to the electrical charge, $n_{j}$ to the number of electrons involved in the $j^{th}$ contribution to the current; F is the Faraday constant ($96,485\;{C\;\mathrm{mol}}^{-1}$), and $n_{Red,j}$ and $n_{Ox,j}$ refer to the number of mols associated with the $j^{th}$ contribution to the electrical current for the reduced or oxidized species, respectively.

### Recording Data and Previous Data Treatment

As we observe the evolution of current during an experiment, we also gather data on other signals, such as changes in mass, absorbance, electromechanical resistance, pH, and more. The time derivatives of these signals compared with the electric current (expressed as dq/dt) [11,35,36,37,38,39,40,41] allow the calculation of ratios, such as F dm/dq or F dA/dq [42,43]. These ratios can be seen as indications of the molar mass of the species involved in the process or the molar absorptivity of the species under examination, respectively.

To do this, the first challenge is ensuring a proper synchronization of data since a small delay between different measured signals can represent a serious problem. In direct current techniques, a small delay between different signals can give a false crossed derivative function. In the case of alternating current, the analysis of unsynchronized signals leads directly to the appearance of artifacts in the impedance response. It must be understood that, in this case, the impedance is already a cross-transfer function between the measured analog signal and the sinusoidal perturbation. In these cases, a pre-calibration is performed to correct the delay between analog signals [43,44,45,46].

Once all signals are correctly synchronized, raw data are usually acquired with a high noise signal. Therefore, the detection of the noisy signal and a previous smoothing data process is mandatory. Some of the routines for this purpose are filters based on the low-pass Fast Fourier Transformations (FFT filter) or Savitzky–Golay (SG) smoothing data [47,48,49], among others. Most of the commercial and free software for plotting data include them. The optimal smoothing procedure depends on the studied system, the recorded signals, and/or the preference of the person analyzing the data. For example, a low-pass FFT filter could be a good choice for removing high-frequency noise. SG procedure could be useful to eliminate a dispersed noise, but the optimum number of points for the SG procedure depends on the number of data points recorded. Whereas poorly filtered signals provide difficult-to-understand derivatives, excessively filtered data leads to derivative curves, which do not correspond to the true measured signal, and smaller and wider peaks appear. There are examples of treated data in the scientific literature [39,42].

After data are correctly synchronized and filtered, there is time to start the crossed derivatives and crossed transfer functions analysis. For that purpose, we revise different coupled electrochemical techniques.

## 3. Spectroelectrochemistry

An oxidation or reduction process is accompanied by changes in the molecule energy levels or the formation/destruction of chemical bonds, and the spectroscopical properties of the molecule also change. These changes can be detected as different bands of light absorption at different wavelengths. On the one side, there is the appearance of new chemical bonds detected as new absorption bands in the IR. However, IR proves difficult to be measured during an electrochemical experiment since the solvent absorption bands can interfere. Part of this problem is solved by measuring the Raman spectra during the electrochemical experiment since water proves transparent to the visible and NIR wavelengths where the LASER Raman usually operates [50,51,52,53]. On the other side, there are also color changes at the vis-NIR or UV that can be easily detected by fast-response modern spectrometers. Spectrometers can work both analogically or digitally (number of counts), measuring light intensity. In the first case, light intensity (*I*) is converted after the appropriate corrections into an absorbance A(λ) measurement, which can be easily related to the concentration of active sites by Lambert–Beer’s law.
(2)$$A\left(\lambda \right)=-\log\left(\frac{I}{I_{0}}\right)=\epsilon \left(\lambda \right)\cdot l\cdot C_{active\;sites}$$
where $I_{0}$ is the light intensity for the blank sample; ϵ(λ) represents the molar absorptivity at the λ wavelength; l is the optical path length, and $C_{active\;sites}$ is the concentration of active sites. In the case of the number of counts (digital spectrometers) and after a previous calibration, light intensity is proportional (∝) to the number of counts. Thus,
(3)$$I\propto n_{counts}\;\;and\;\;I_{0}\propto n_{counts,0}$$
where $n_{counts}$ is the number of counts registered for the sample at a given wavelength (λ), and $n_{counts,0}$ the number of counts for the blank at this wavelength.
(4)$$A\left(\lambda \right)=-\log\left(\frac{I}{I_{0}}\right)=-\log\left(\frac{n_{counts}}{n_{counts,0}}\right)\;$$

The UV-vis-NIR diode array spectrometers proved to be of special interest for our purposes since they allow recording of one spectrum in the wavelength range from 350 to 1000 nm instantaneously.

### 3.1. Spectroelectrochemistry in the dc-Regime

In the case of studying a redox process of such adsorbed species as electrogenerated polymers, recording the spectrum of the reduced and oxidized forms can help to identify the changes that have occurred in the molecule (i.e., the formation of polarons or bipolarons). In addition, the evolution of light signals during the electrochemical experiment gives extra kinetic information. Let us assume a reduction reaction taking place by the general reaction:(5)$$Ox+ne^{-}⇄Red$$

The current at any time is directly related to the kinetics of the redox process.
(6)$$i\left(t\right)=\frac{dq\left(t\right)}{dt}=nFA\frac{d\Gamma_{Ox}}{dt}=-nFA\frac{d\Gamma_{Red}}{dt}$$

In Equation (6), $\Gamma_{Red}$ and $\Gamma_{Ox}$ refer to the surface concentrations of the reduced and oxidized species, respectively. The rest of the parameters have their usual meaning. If absorbance at a given wavelength is related to the concentration of active sites,
(7)$$A\left(\lambda \right)=\varepsilon_{Ox}\left(\lambda \right)lC_{Ox}+\varepsilon_{Red}\left(\lambda \right)lC_{Red}=\varepsilon_{Ox}\left(\lambda \right)\Gamma_{Ox}+\varepsilon_{Red}\left(\lambda \right)\Gamma_{Red}$$

It should be noted that the surface concentration (Γ) is given by the product of layer thickness (l) and the volume concentration (C). If the total number of active centers at any time does not change during the electrochemical process or, in other words:(8)$$\Gamma_{Ox}\left(t\right)+\Gamma_{Red}\left(t\right)=\Gamma_{total}$$

Differentiating the Equation (7), we arrive at
(9)$$\frac{dA\left(\lambda \right)}{dt}=\varepsilon_{Ox}\left(\lambda \right)\frac{d\Gamma_{Ox}\left(t\right)}{dt}+\varepsilon_{Red}\left(\lambda \right)\frac{d\Gamma_{Red}\left(t\right)}{dt}$$
and simplifying,
(10)$$\frac{dA\left(\lambda \right)}{dt}=\left(\varepsilon_{Red}\left(\lambda \right)-\varepsilon_{Ox}\left(\lambda \right)\right)\frac{d\Gamma_{Red}\left(t\right)}{dt}=-\left(\varepsilon_{Red}\left(\lambda \right)-\varepsilon_{Ox}\left(\lambda \right)\right)\frac{d\Gamma_{Ox}\left(t\right)}{dt}$$

Even if molar absorptivities and the cell length were unknown, the time derivative of the absorbance proves proportional to the time derivative of the concentration of active sites (reduced or oxidized).
(11)$$\frac{dA\left(\lambda \right)}{dt}=k\left(\lambda \right)\frac{d\Gamma_{Red}\left(t\right)}{dt}=-k\left(\lambda \right)\frac{d\Gamma_{Ox}\left(t\right)}{dt}$$

In Equation (11), k(λ) is the difference between the molar absorptivities of the reduced and the oxidized forms. Looking at the great similitude between current (Equation (6)) and the time derivative of absorbance (Equation (11)), if there is only one electrochemical process causing color changes at a given wavelength, the shape of the time derivative of absorbance should be similar to the shape of the current curve [54]. In addition, the time derivative absorbance can solve one of the most important issues in spectroelectrochemistry—absorbance needs a previous calibration (blank measure) at each wavelength, and this is not always an easy experimental task. The use of time derivative absorbance can solve this issue. If we differentiate Equation (2) or (4), we have
(12)$$\frac{dA\left(t\right)}{dt}=-\frac{d\left(log\left(I/I_{0}\right)\right)}{dt}=-\frac{dlog\left(I\right)}{dt}+\frac{dlog\left(I_{0}\right)}{dt}=-\frac{dlog\left(I\right)}{dt}$$

Since $\log\left(I_{0}\right)$ does not change with time, there is no need to measure the blank sample for the derivative analysis.

By the combination of both Equations (6) and (11), we can have the FdA(λ)/dq function as a first estimation of the electrochromic efficiency at each wavelength but also at each applied potential during the electrochemical process [42]:(13)$$F\frac{dA\left(\lambda \right)}{dq}=F\frac{\frac{dA\left(\lambda \right)}{dt}}{i\left(t\right)}=-\frac{k\left(\lambda \right)}{nA}$$

The parameter k(λ) is characteristic for the electrochemical reaction studied at each wavelength. The sign of k(λ) is given by the difference between the molar absorptivities of the reduced and oxidized forms. If there is only one electrochemical process associated with one absorbance change at a given wavelength, the crossed derivative function $F\frac{dA\left(\lambda \right)}{dq}$ gives a measure of the electrochromic efficiency (difference between molar absorptivities) for this process [42]. It should be noted that this parameter cannot be evaluated in the regions of the voltammogram where the current is close to 0 since the obtained asymptotic values cannot be interpreted.

Even if more than one process takes place simultaneously, it is possible to deconvolute the overall response. Let us assume that the global current measured during the experiment can be expressed as the sum of currents due to different individual processes:(14)$$i\left(t\right)=\underset{j}{\sum }n_{j}FA\frac{d\Gamma_{j,Ox}}{dt}=\underset{j}{\sum }-n_{j}FA\frac{d\Gamma_{j,Red}}{dt}$$

This expression for the voltammetry of adsorbed species case can be rewritten as
(15)$$i\left(t\right)=\underset{j}{\sum }\frac{n_{j}FA\Gamma_{Total}bv\cdot e^{b\left(E_{s}+vt-E_{j}^{0’}\right)}}{{\left(1+e^{b\left(E_{s}+vt-E_{j}^{0’}\right)}\right)}^{2}}=\underset{j}{\sum }\;\frac{i_{m.j}}{\cosh^{2}\left(\frac{b}{2}\left(E_{s}+vt-E_{j}^{0^{\prime }}\right)\right)}$$

In Equation (15), b=F/RT; v refers to the scan rate, $E_{s}$ to the starting potential, and $E_{j}^{0^{\prime }}$ to the formal potential of the electrochemical process. $i_{m,j}$ is the maximum current for the $j^{th}$ electrochemical process
(16)$$i_{m,j}=\frac{n_{j}FA\Gamma_{Total}bv}{4}$$

Moreover, the change in absorbance measured at a characteristic wavelength is given by the sum of time derivatives of absorbances associated with each individual process:(17)$$\frac{dA\left(\lambda \right)}{dt}=\underset{j}{\sum }k_{j}\left(\lambda \right)\frac{d\Gamma_{j,Red}\left(t\right)}{dt}=\underset{j}{\sum }-k_{j}\left(\lambda \right)\frac{d\Gamma_{j,Ox}\left(t\right)}{dt}$$

If each one of the electrochemical processes causes absorbance changes at different wavelengths, then it is possible to assign a singular characteristic wavelength for each one of these processes, and thus, to separate the electrochemical response for each one of the overlapped processes. This procedure has been successfully used for the deconvolution of electrochemical processes during the redox processes in poly (Azure A) films [42]. In this case, electrochromic efficiencies were estimated at different wavelengths from the $F\frac{dA\left(\lambda \right)}{dq}$ function obtained during the voltammetric curve, Equation (13), and then associated with individual electrochemical processes. Therefore, in most cases, there is a characteristic wavelength where electrochromic efficiency proves optimal to follow each one of the overlapped electrochemical processes.

It is also possible that one of the electrochemical processes does not cause absorbance changes at any λ; for example, an electrocatalytic process, such as hydrogen evolution. That causes a fast decrease in the absolute value of the electrochromic efficiency since there is a part of the current that does not cause any absorbance change. Thus, the time derivative of the absorbance shows a zero baseline, but the current can show a non-zero (increasing or decreasing) baseline due to the electrocatalytic reaction [55].

Figure 1 presents a simulation of a voltammetric curve for the reduction in adsorbed species, where two electroactive sites are present. Each electroactive center is associated with a voltammetric peak, and each electrochemical reaction causes changes in color; one of them causes positive changes in absorbance, but the other causes negative ones. Together with this “ideal” situation, a catalytic current is added to the measured current, which causes a clear decrease in the electrochromic efficiency at the smaller potential.

If the material electrodeposited is a conducting polymer, the absorbance due to the organic polymers generated on electrode’s surface is dependent on the thickness of electrogenerated polymers, and, thus, by absorbance changes, it is possible to estimate the amount of the electrogenerated polymer on transparent or reflective electrodes [37]. The main advantage is that kinetics for the polymer growth can be followed by measuring absorbance at different wavelengths. However, some handicaps should be considered. One of these is to work when the film is thick enough, and the absorbance does not change even if the electrodeposit continues. Another obstacle is that we need to work in very diluted monomer solutions since, at high concentrations, the monomer solution could be very dark, and the absorbance could be saturated.

Spectroelectrochemistry can also be of interest for the study of electrodeposition, electrodissolution, or the growth of a passive layer on the surface of a metal. In those cases, the analysis of the spectroscopic data should be considered only when the electrode’s surface changes by the new metal layer electrodeposited since, if the metal growths on the surface of the previously electrodeposited metal, there is no color change and absorbance keeps constant. Another point of interest is the electrodissolution of a metal where some intermediate and colored species are formed near the electrode’s surface that can be detected by changes in absorbance/color at a given wavelength. This information proves very interesting for the kinetic mechanism studies since it allows the detection of intermediate species by their color at different wavelengths. If there is enough absorbance resolution from the kinetics of the appearance/disappearance of the intermediate species, it is possible to elucidate the kinetics constants for consecutive reactions [51,56].

### 3.2. Spectroelectrochemistry in ac-Regime

Impedance techniques allow obtaining information separately from fast and slow processes during an electrochemical experiment by changing the frequency of the perturbation. Then, at higher frequencies, there is only the response due to the faster processes, while at the lower frequencies, faster and slower processes contribute to the general response. Coupling electrochemical impedance spectroscopy (EIS) techniques with a fast spectrometer working at a fixed wavelength could serve to analyze the sinusoidal absorbance response to the perturbation and then obtain the “color” impedance spectroscopy (CIS). Impedance is measured with the help of the frequency response analyzer (FRA) that compares both the sinusoidal potential perturbation and the sinusoidal current response, obtaining the impedance transfer function $\frac{\Delta E}{\Delta i}\left(\omega \right)$ [19,25,28,43]. For the simplest case of only one electrochemical reaction, this function can be converted into the electric charge transfer function at each frequency via
(18)$$\frac{\Delta q}{\Delta E}\left(\omega \right)=\frac{1}{\frac{j\omega \Delta E}{\Delta i}}=-nFA\frac{{\Delta \Gamma }_{red}}{\Delta E}\left(\omega \right)$$

For the color impedance, the transfer function is given by the following:(19)$$\frac{\Delta A\left(\lambda \right)}{\Delta E}\left(\omega \right)=k\left(\lambda \right)\frac{{\Delta \Gamma }_{red}}{\Delta E}\left(\omega \right)$$

In these expressions, ω refers to the angular frequency of the sinusoidal potential perturbation (ΔE); j is the imaginary unit ($\sqrt{-1}$), and Δi is the current response to the potential perturbation ΔE. ΔA is the response in absorbance change in the sample to the potential perturbation ΔE.

Therefore, and from Equations (18) and (19), it is possible to get an estimation of the electrochromic efficiency k(λ). Here, the main advantage is that it is possible to measure this efficiency at different frequencies (ω). As an example of this kind of analysis, if the electrochemical process causing the color change takes place simultaneously with another electrochemical process not causing color changes but is slower, then at the higher frequencies, a large value for the electrochromic efficiency is measured, while at the lower frequencies, the smaller values are detected.

The frequency response analyzer (FRA) analyzes current, potential, and analog, corresponding to the current measured in a photodetector proportional to the light intensity at a given wavelength. Therefore, the transfer function measured corresponds to the $\frac{\Delta I\left(\lambda \right)}{\Delta E}\left(\omega \right)$ and not to the $\frac{\Delta A\left(\lambda \right)}{\Delta E}\left(\omega \right)$ transfer function. However, for small potential amplitude perturbation, there is a direct relation between both transfer functions [43]:(20)$$\frac{\Delta A\left(\lambda \right)}{\Delta E}\left(\omega \right)\approx B\frac{\Delta I\left(\lambda \right)}{\;\Delta E}\left(\omega \right)$$
where B proves constant at a given wavelength. Then, the measured transfer function can be interpreted as the $\frac{\Delta A\left(\lambda \right)}{\Delta E}$ transfer function.

For the case of adsorbed electroactive species, if we assume a fast electron transfer at the electrode/solution interface and that adsorbed species form a thin layer on the electrode, then the concentration of one electroactive species at the electrode can change according to [43]
(21)$$\frac{{\Delta \Gamma }_{i}}{\Delta E}\left(\omega \right)=\frac{-G_{i}}{\;1+\frac{K_{i}}{d}\frac{\cot h\left(\sqrt{j\omega \tau_{i}}\right)}{\sqrt{j\omega \tau_{i}}}}\left(\omega \right)$$
where ${\Delta \Gamma }_{i}$ refers to the oscillating concentration of the electroactive species; $G_{i}$ and $K_{i}$ are the parameters related to the kinetic constants for the electrochemical reaction and the concentrations of electroactive species [43]; d refers to the thickness of the electroactive layer, and $\tau_{i}$ to the time constant for the $i^{th}$ electrochemical process. Equation (21) proves that it is easy to relate color changes (color impedance) with the electrical charge transfer functions (Δq/ΔE) and the color impedance transfer functions (ΔA(λ)/ΔE).
(22)$$\frac{\Delta q}{\Delta E}\left(\omega \right)=\underset{i}{\sum }\frac{1}{j\omega }\frac{FG_{i}}{1+\frac{K_{i}}{d}\frac{\coth\left(\sqrt{j\omega \tau_{i}}\right)}{\sqrt{j\omega \tau_{i}}}}$$
(23)$$\frac{\Delta A\left(\lambda \right)}{\Delta E}\left(\omega \right)=\underset{i}{\sum }\frac{1}{j\omega }\frac{k_{i}\left(\lambda \right)G_{i}}{1+\frac{K_{i}}{d}\frac{\coth\left(\sqrt{j\omega \tau_{i}}\right)}{\sqrt{j\omega \tau_{i}}}}$$

Figure 2 shows a simulation of how a color impedance spectrum looks in a system where two electroactive centers cause color changes at the same wavelength. At the higher frequencies, there is only the response of the faster process (the higher frequency loop), while at the lower frequencies, there is the sum of both contributions. It should be noted that both simulated spectra will look identical to the electric charge transfer function, but color impedance allows to separate both contributions by their different electrochromic efficiencies. In Prussian Blue films studied by color impedance, it has been possible to identify different electroactive centers by their different electrochromic efficiency $k_{i}\left(\lambda \right)$ at 690 nm and 1000 nm of wavelength [43,57]. A general problem for all impedance studies is that they require the system to be in a steady state to obtain a good stable response, especially for the lower frequencies. In the case of unsteady systems, this can result in artifacts that make it impossible to accurately determine the electrochromic properties of the material under study.

## 4. Electrogravimetry

As in the above-mentioned color impedance, electrogravimetry consists in coupling electrochemical techniques with gravimetric ones. Since electrochemical processes are heterogeneous processes, there is always a mass change on the electrode’s surface due to adsorption/desorption processes, electrodeposition, corrosion, charge balance during redox processes of adsorbed species, etc. This is why electrogravimetry has been used for the study of kinetics during electrodeposition processes [55], electrodissolution processes [10,15,39], the formation of passive layers on the metal electrode’s surface [15], adsorption/desorption of active substances during electrochemical processes [58], or for the study of charge transfer reaction in conducting polymers [5,43]. Changes in mass can be indirectly recorded during the experiment thanks to the quartz crystal microbalance. This device measures the resonance frequency of a quartz crystal, and shifts of this frequency are converted into mass changes with Sauerbrey’s equation [1]. This equation applies to thin layers and small changes in mass and provides that the viscoelastic properties at the electrode/dissolution interface do not dampen the changes in resonance frequency.

If all these conditions apply, there is a measure of mass changes on the electrode’s surface during an electrochemical experiment. Mass gives excellent information to distinguish among different kinetic mechanisms if accurately measured and synchronized with the electrochemical response. For that purpose, the first step is a calibration procedure where an experimental Sauerbrey’s constant is estimated from a stoichiometric electrochemical reaction. Usually, the electrodeposition of copper or silver is considered a good calibration method [59,60]. Once the calibration constant is obtained, mass changes are obtained from frequency shifts detecting changes of about 10 ng.

We can see some cases where electrogravimetry has proven useful for the characterization of electrochemical processes.

### 4.1. Conducting Polymers. The Insertion Problem

Conducting polymers have been revealed as one of the most studied systems due to their possibilities in different electrochemical applications, going from electrocatalysis to their possible use in electrical charge storage devices. If one polymer changes its oxidation state, there is an excess or lack of electric charge within that should be balanced. The insertion model schematizes this problem for the cation insertion/expulsion as follows:(24)$$\langle \;P\;\rangle +ne^{-}+\frac{n}{z^{+}}M^{z^{+}}⇄\;\langle \;P,ne^{-},\frac{n}{z^{+}}M^{z^{+}}\rangle$$

In many cases, the electric charge balance takes place during the exchange of anions, and then,
(25)$$\;\langle \;P,\frac{n}{z^{-}}A^{z-\;}\rangle +ne^{-}⇄\;\langle \;P,ne^{-}\;\rangle +\frac{n}{z^{-}}A^{z-}$$

In these expressions, 〈 P 〉 refers to the polymer matrix; $M^{z^{+}}$ refers to the cation with electrical charge $z^{+}$, and $A^{z^{-}}$ to the anion with electrical charge $z^{-}$; n is the number of electrons exchanged.

In both cases, there are mass changes associated with the electrochemical process if the polymer is attached to the electrode’s surface. If the electric charge balance takes place during the exchange of anions, during the reduction process, there is a mass decrease in the electrode’s surface, and the opposite happens during the oxidation process. For the cations exchange case, mass increases during the reduction and decreases during the oxidation. In order to make a better comparison between current and mass changes, the time derivative mass functions are preferred. If several processes take place simultaneously, each one of them with a characteristic molar mass for the exchanged species ($MW_{i}$), the time derivative of mass can be estimated via
(26)$$\frac{dm\left(t\right)}{dt}=-\frac{1}{F}\underset{i}{\sum }\frac{MW_{i}}{z_{i}}\frac{dq_{i}\left(t\right)}{dt}=-\frac{1}{F}\underset{i}{\sum }\frac{MW_{i}}{z_{i}}\nu_{i}\frac{dq\left(t\right)}{dt}$$
where $z_{i}$ represents the electrical charge of the ion i that participates in the electrochemical process, and $\nu_{i}\frac{dq\left(t\right)}{dt}$ is the associated charge compensated by the participation of each *i* species. Equation (26) implies positive values for anions’ participation during the oxidation process and negative during the cations process.

If mass changes are related to the electrochemical response, we can assess the species participating in the electrochemical reactions at a given potential by calculating F(dm/dq) from dm/dt (in g s^−1^) and current (in A) data as [11,39]:(27)$$F\frac{dm}{dq}=F\;\frac{dm\left(t\right)/dt}{dq\left(t\right)/dt}=\sum^{\;}\nu_{i}\frac{MW_{i}}{-z_{i}}\pm \xi$$

Fdm/dq can be interpreted as a mean weighted value of the molar masses of exchanged species during the electrochemical process. It should be noted that anions and cations contribute differently to this molar mass, and cations’ contribution can cancel the anions’.

In Equation (22), ξ corresponds to the mass contribution of solvent or non-charged species. In an ideal system considering only charged species, negative F(dm/dq) means that the electroactive system exchanges cations (dm/dt < 0 and dq/dt > 0 or vice versa), whereas a positive F(dm/dq) indicates the exchange of anions (dm/dt > 0 and dq/dt > 0 or vice versa). In real systems, the expected F(dm/dq) value for the charged participating species is subject to modification due to some factors, such as the transfer of neutral molecules involving only mass change and the simultaneous participation of different species, if there are surface catalytic reactions or capacitive currents that only consume a charge. In the last case, F(dm/dq) tends to be 0 g mol^−1^. One limitation of this analysis is that mathematically, F(dm/dq) tends to be infinite if the current approaches zero.

This function offers a quick estimation of the molar mass of exchanged species. If there is only one process balanced by the participation of one ion, this function gives the molar mass of the exchanged species divided by $z_{i}$. If more than one species participates or there is more than one electrochemical reaction, this function gives a weighted mean molar mass of the exchanged species but considering their sign (positive or negative).

This methodology has been used to identify the cation or anion inserted during the electrochemical processes of different conducting polymers allowing to quantify the number of molecules that accompany the cations or anions during its exchange, as well as solvating the ion or by the exclusion effect. Since the electric charge for the solvent molecules is 0, it introduces an increase in the absolute value of $F\frac{dm}{dq}$, if their flux is accompanying the ion or a decrease, if their flux is opposite to the ions flux (parameter ξ in Equation (27) [39]).

Looking at the simulation of Figure 3, one can see how the time derivative mass shows two peaks at the same potentials as the current peak potentials. At the more negative potentials, there is the electrocatalytic process which causes a small increase in the absolute value of the current but no mass change. This is reflected in the Fdm/dq evolution at these potentials, showing a near-to-zero value. This is an example of how this analysis can detect the presence of catalytic processes during a voltammetric experiment.

Some of the limitations of this way of working are that mass changes are associated with changes in the resonance frequency of the EQCM electrode. However, for this relationship to exist, some conditions must be satisfied, such as that the layer on the microbalance electrode should be sufficiently rigid. An additional difficulty is a change in the viscoelastic properties or thickness of the conducting polymers, which can also cause small shifts in the resonance frequency not associated with mass changes. Working with thick polymer layers poses a problem in that the linearity between mass and resonance frequency changes is lost, and the mass measurement is not valid.

### 4.2. Corrosion and Formation of Passive Layers

As commented above, EQCM proves to be of special interest in all these processes involving mass changes. Kinetics for corrosion reactions is one of the most studied processes in electrochemistry and engineering. Corrosion rates are usually expressed in mg per year. Therefore, one can think that microbalance coupled with the electrochemical studies of corrosion is an excellent option. However, the corrosion mechanism is more complex than a loss of mass during a certain time. Depending on the experimental conditions, a corrosion process implies a mass decrease in the studied sample; however, in some cases, if the oxidation subproducts keep on the metal surface, there is a mass increase. The presence of different anions, such as $Cl^{-}$, can accelerate the corrosion process. The role of these anions involves previous adsorption on the metal that can help to stabilize some complexes with the metal accelerating the corrosion rate. In other cases, the presence of some anions can help the stabilization of $M{\left(I\right)}_{aq}$ species making the electrodissolution process faster. In all these cases, the EQCM can help interpret experimental results.

Let us propose a mechanism for the dissolution of metal in two single electron steps, followed by a transport stage of the oxidized species towards the solution [11] Equations (28)–(30):(28)$$M\left(s\right)\to M{\left(I\right)}_{ads}+e^{-}$$
(29)$$M{\left(I\right)}_{ads}\to M{\left(II\right)}_{ads}+e^{-}$$
(30)$$M{\left(II\right)}_{ads}\to M{\left(II\right)}_{aq}$$

In these expressions, the role of different anions stabilizing the oxidized metal species is not considered since, globally, the anions do not keep on the electrode’s surface after the metal leaves the electrode’s surface. Therefore, the M(II) species leave the metal surface after losing two electrons, and the $F\frac{dm}{dq}$ crossed-derivative function gives a value of $-\frac{MW_{M}}{2}$. If we consider that anions can stabilize the M(I) species, helping to remove it before the second electron transfer takes place,
(31)$$M\left(s\right)\to M{\left(I\right)}_{ads}+e^{-}$$
(32)$$M{\left(I\right)}_{ads}+xA^{-}\to {\left(MA\right)}_{aq}^{x-1}\;$$

In this case, the $F\frac{dm}{dq}$ crossed-derivative function will reach values of $-\frac{MW_{M}}{1}$ since only one electron is needed to remove the metal from the electrode’s surface.

This methodology has been used for the kinetics study of electrodissolution of Zn, Ni, and Cu in sulfate and chloride solutions. In the case of Cu electrodissolution, the Cu(I) species were stabilized on the electrode’s surface, and a first increase in mass was detected during the electrodissolution [39]. These mechanisms were tested by different electrochemical techniques, such as EIS, obtaining experimental results that corroborate these hypotheses [11,15].

Another kind of studies that could be followed by the coupling of EQCM and electrochemical techniques is the growth of passive layers during the oxidation of a metal [15]. In this case, the corrosion of a metal does not give a decrease in mass since a layer of oxides/hydroxides or other insoluble salts of the metal can be formed on the metal surface. Thus, even during a corrosion experiment, mass increases. The Point Defect Model [61,62] is used for the interpretation of the electrochemical response during the growth of the passive layer, and the EQCM gives the rate of mass increase. In this case, the use of the $F\frac{dm}{dq}$ function allows to identify the nature of the species formed on the electrode’s surface during the growth of the passive layer and the kinetics of a passive layer growth.

The application of the quartz microbalance to this type of studies has some limitations. Firstly, it is necessary to work with quartz microbalance electrodes, usually gold, and this implies that the metal must be electrodeposited on this electrode. On the other hand, and as a limitation of the EQCM, it is not possible to work with very thick layers of metal.

### 4.3. Electroplating and Electrodeposition

Electrodeposition is used to modify the surface of one electrode and its properties by the generation of different substances on its surface. It is possible to distinguish between at least two kinds of electrodeposits. On the one hand, there is electroplating, where a metal layer covers the electrode’s surface; on the other hand, there is the electrode position of organic layers on the surface of a conducting electrode. Here, we can consider the generation of an intrinsically conducting polymer as well as the formation of an isolating material.

In the case of electroplating, usually, this is a process taking place under galvanostatic conditions when the rate of growth should be constant. However, experimental conditions for the electrodeposition require the application of enough negative potentials, and then, undesirable parallel electrochemical reactions, such as the hydrogen evolution reaction (HER), could take place. If the current is only measured during the electrodeposition reaction, it proves impossible to estimate the electric efficiency for the electrodeposition reaction. However, if the mass is simultaneously recorded, one can have a fast estimation of the electrodeposition reaction efficiency by measuring the $F\frac{dm}{dq}$ function. If electrodeposition takes place by the reaction
(33)$$M_{aq}^{z+}+ze^{-}\to M_{ads}$$
absolute values of $F\frac{dm}{dq}$ smaller than $\frac{MW_{M}}{z}$ mean that a part of the electric charge is wasted in the HER or other coupled electrochemical processes. As a case of the use of the Fdm/dq function to characterize the electrochemical properties of some materials, we will consider the electrodeposition of a metal (Zn) that takes place together with the hydrogen evolution reaction (HER). In this case, we assume that the current is the sum of the two contributions, but the mass changes are only associated with the metal deposition process. Thus, we can write the experimental variation of Fdm/dq as
(34)$${F\frac{dm}{dq}|}_{exp}=\nu_{1}F{\frac{dm}{dq}|}_{1}+\left(1-\nu_{2}\right)F{\frac{dm}{dq}|}_{2}=\nu_{1}F{\frac{dm}{dq}|}_{1}+0$$

In Equation (34) we have the parameter $\nu_{1}$, which represents the part of the current that we associate with the first process, and, therefore, $\left(1-\nu_{1}\right)$ corresponds to the current associated with the second process. ${F\frac{dm}{dq}|}_{1}$ and $F{\frac{dm}{dq}|}_{2}$ are the expected values for the function Fdm/dq, according to processes 1 or 2, respectively. In this case, ${F\frac{dm}{dq}|}_{1}=-\frac{M\left(Zn\right)}{2}=-32.7{\;g\;\mathrm{mol}}^{-1};$ the mass of the metal divided by its charge and for the second process ${F\frac{dm}{dq}|}_{2}=0$ since this is a reaction taking place without mass change. By solving this equation for each potential, we can separate the current for each process and, thus, isolate the catalytic current. For the case of Zn electrodeposition, a value of $\frac{Fdm}{dq}=-23\;{g\;\mathrm{mol}}^{-1}$ was obtained during the deposition of the metal in these experimental conditions [11], which corresponds to the consideration that for each atom of Zn deposited, ½ of $H_{2}$ gas is formed.

This way of working can be generalized to other cases when the two values of Fdm/dq are different, without necessarily canceling one of them, and different contributions to the overall current can be separated.

Therefore, it could be possible to optimize the experimental conditions (controlled current, additives, temperature, pH, and concentration) to achieve the largest electric efficiency. The electrodeposition of Cu from a ${\mathrm{CuSO}}_{4}$ aqueous and $H_{2}{\mathrm{SO}}_{4}$ solution has been used for the calibration procedure of EQCM devices [59]. The redox potential for ${\mathrm{Cu}}^{2+}/\mathrm{Cu}$ is 0.34 V, while for the ${\mathrm{Cu}}^{2+}/{\mathrm{Cu}}^{+}$, it is 0.16 V. Then, during electroplating, the most favorable reaction is the reduction of ${\mathrm{Cu}}^{2+}$ to Cu, and $F\frac{dm}{dq}$ reaches constant values near $\frac{65}{2}{\;g\;\mathrm{mol}}^{-1}$. However, if the same experiment is made with $Cl^{-}$ anions in the solution, since chloride can stabilize ${\mathrm{Cu}}_{aq}^{+}$ species, the $F\frac{dm}{dq}$ measured does not correspond exactly to the $65/2{\;g\;\mathrm{mol}}^{-1}$ [39].

As commented above, this methodology has also been used for the characterization of the electrodeposition of conducting polymers on the EQCM electrodes. In this case, electrodeposition can take place by following different strategies. On the one side, there is the case of organic conducting polymers, such as those generated from phenothiazine monomers, that need the formation of a radical cation to start the polymerization process. After the radical cation has been formed, the monomer unit bonds with the polymer chain. Obviously, there is a mass increase during the electrodeposition due to the new monomers attached to the polymer chain, but usually, the formation of radical cations requires large potentials that also can cause the oxidation of the polymer. Therefore, changes in mass due to the charge in balance associated with the oxidation process should also be considered for a correct interpretation. In this case, both the polymer chain growth and the polymer oxidation reaction contribute to the $F\frac{dm}{dq}$ function [55]. If the charge balance takes place during the exchange of anions, mass increases more than expected for a single polymer chain growth; on the contrary, if the polymer exchanges cations during the oxidation process, a smaller mass change should be expected. In any case, the polymerization process needs the formation of more than one radical cation by each monomer attached to the polymer, making the efficiency to be always smaller than 100%. The use of $F\frac{dm}{dq}$ during the growth of conducting polymers can help in optimizing the experimental conditions for the largest electric efficiency.

### 4.4. Electrogravimetry in the ac-Regime

As in the case of spectroelectrochemistry, it is also possible to record mass impedance together with electrochemical impedance spectroscopy. This ensemble of techniques is known as *ac*-electrogravimetry and was first introduced by the Gabrielli and Keddam research group [44]. The interpretation of *ac*-electrogravimetry data requires an accurate data treatment. First, as mass impedance is recorded from resonance frequency shifts, and this measurement takes a little more than current or potential measurement, a calibration–correction procedure is needed to correct this delay. After this, the *ac*-electrogravimetric transfer function is obtained. If there are several electrochemical processes causing mass changes on the electrode’s surface, and each of these processes is characterized by the molar mass of the exchanged species, then the transfer function can be expressed as [43]
(35)$$\frac{\Delta m}{\Delta E}\left(\omega \right)=-\frac{1}{F}\underset{i}{\sum }\frac{MW_{i}}{z_{i}}\frac{\Delta q_{i}}{\Delta E}\left(\omega \right)=-\underset{i}{\sum }\frac{1}{j\omega }\frac{MW_{i}/z_{i}{}_{i}\cdot \;G_{i}}{1+\frac{K_{i}}{d}\frac{\coth\left(\sqrt{j\omega \tau_{i}}\right)}{\sqrt{j\omega \tau_{i}}}}$$

In the case of an exchange of cations/anions in a conducting polymer, this transfer functions produces loops in different quadrants on the complex plane of mass impedance.

Figure 4 shows simulations of the mass transfer function for different cases. For the cations exchange, a third quadrant semicircle is observed, while for the anions, on the first quadrant. If there is more than one participating ion, there is a transition from high to low frequencies, starting on the quadrant corresponding to the fast exchange process and changing at the lower frequencies to the quadrant corresponding to the slowest process [43,63,64].

From Equation (35), we can arrive at the crossed impedance transfer function $F\frac{\Delta m}{\Delta q}\left(\omega \right)$
(36)$$F\frac{\Delta m}{\Delta q}\left(\omega \right)=F\frac{\Delta m/\Delta E}{\Delta q/\Delta E}\left(\omega \right)=-\underset{i}{\sum }\frac{MW_{i}}{z_{i}}\nu_{i}$$

This transfer function can be interpreted as the molar mass divided by its electric charge of the exchanged ion if only one species is exchanged. In this case, the imaginary part equals to 0, and the real part, to $MW_{i}/z_{i}$. If more than one ion participates, this is a mean and weighted molar mass of the exchanged species, but in the complex plane. Differently shaped loops can be observed depending on the relative speed of the electrochemical processes studied. Mass changes can also be caused by the exchange of non-charged (solvent) species. If these species are exchanged accompanying the cation/anion, then the apparent molar mass of the ion is corrected by this contribution. If this exchange of solvent takes place at a different rate, it will be detected in this transfer function as a different process in mass changes, but not in electric charge, and a loop will appear in the $F\frac{\Delta m}{\Delta q}\left(\omega \right)$ transfer function [43]. As an example, during the electrochemical characterization of PEDOT films by *ac*-electrogravimetry (Figure 5), the value of this transfer function at different frequencies has allowed identifying the exchanged species during the charge compensation process in PEDOT films at different potentials [55]. At the higher frequencies (0.1 Hz–10 Hz), there is the exchange of $\mathrm{Li}{\left(H_{2}O\right)}_{x}^{+}$ since the mass proves larger than the ${\mathrm{Li}}^{+}$ cation, but at the lower frequencies, the slow electrochemical reduction in molecular $O_{2}$ causes a larger current but no mass change, and then, FΔm/Δq decreases.

It has also been possible to discern between the fast and slow cation exchange in Prussian Blue films at different potentials [43,57]. In other studies, this transfer function has served to evaluate the number of water molecules accompanying cations or to exclude the insertion of anions within the films [65,66].

Despite the great possibilities offered by this technique, there are limitations. In the first place, there are limitations derived from the use of microbalances, such as stiffness of the films, thin films, changes in viscoelastic properties, etc. In addition, it is not possible to work with a conventional microbalance, but it is necessary to use devices that allow a faster response to frequency changes, and this requires previous calibration.

### 4.5. Electromechanical Properties at the Electrode Interface

Electrochemical techniques are always referred to as heterogeneous processes. There is an electron-conducting electrode and an ionic conductor in contact with it. Electrochemical reactions take place on the electrode’s surface, but what happens at the electrode/solution interface? It proves difficult to characterize the interface region during an electrochemical process. This interface region should be interpreted as a transition region from the solid/liquid electrode to the solution (or ionic conductor). Obviously, this interface region changes during the electrochemical reaction, but how can we characterize these changes? There is the possibility to characterize this interface region by LASER beam deflection, mirage effect, or other spectroscopical techniques [67,68,69], where changes in the density at the interface region during the electrochemical experiment can serve to estimate the flux of species.

There is also the possibility to use the quartz crystal microbalance to follow these changes. The resonator electrode is modeled as the Butterworth–Van Dyke (BVD) equivalent circuit [70]. The simplest BVD-based circuit consists of two branches that represent a static capacitance ($C_{s}$) in parallel with a motional branch with the motional inductance $L_{m}$, the motional capacitance $C_{m}$, and the motional resistance $R_{m}$ in series. Particularly, $R_{m}$ describes the energy dissipation during oscillation caused by internal frictions, mechanical losses, and acoustical losses to the adjacent environment (solution) [71,72]. It is true that the resonant frequency depends slightly on changes at the interface region, but this is not the case for the motional resistance $R_{m}$. This resistance is measured with some EQCM devices to correct frequency shifts if the electrode interface consists of a viscous material [72]. There are different models that can relate to this motional resistance and its variation with the viscoelastic properties on the electrode material, but also with the properties of the interface region, such as magnetic properties, molecular interactions, film porosity, metal conversions, or electrocatalysis [34,72,73,74,75,76,77,78].

The use of this magnitude for the characterization of electrochemical properties of different kinds of films is relatively recent. For that, it was necessary to consider that the acoustic wave generated by the quartz resonator crosses some physical barriers in the normal direction of the resonator: the quartz crystal together with the metal electrode (q); the quartz|film interface (q|f); the acoustically thin film (f); the film|solution interface (f|s); and finally, the acoustic wave is damped in the liquid solution (s). Therefore, $R_{m}$ can be expressed as a sum of energy dissipations, such as [74,79,80,81]:(37)$$R_{m}=R_{m}\left(q\right)+R_{m}\left(q|f\right)+R_{m}\left(f\right)+R_{m}\left(f|s\right)+R_{m}\left(s\right)$$
where $R_{m}\left(q\right)$ and $R_{m}\left(s\right)$ are assumed constant during the experimental time transient and can be eliminated as in
(38)$$\frac{dR_{m}}{dt}=\frac{d\left(R_{m}\left(q|f\right)+R_{m}\left(f\right)+R_{m}\left(f|s\right)\right)}{dt}$$

Thus, the evolution of Equation (38) is only due to such film physic properties as the interaction forces between the electrodes surface and the film, the film roughness, which leads the friction between the film surface and the liquid, and the viscoelasticity of film due to the creation and destruction of covalent bonds by electrochemical reactions [82,83].

In other works, changes in the motional resistance are used to explain the mechanism of formation of the passive layer on the electrode’s surface, to calculate the thickness of viscoelastic films, or to monitor the coupling of biomolecules [84,85]. There is also the possibility to relate these parameters with the viscosity or surface tension of different fluids in contact with the EQCM electrode [79]. In 1995, Muramatsu et al. analyzed the Sauerbrey model through the relationship between $∆R_{m}$ and $∆f_{r}$ of viscoelastic films and defined $df_{r}/dR_{m}$ as the energy dissipation factor or viscoelasticity factor of coated film [86,87,88]. When $∆f_{r}/∆R_{m}$ ratios are equal or greater than 100 Hz Ω^−1^, the relation between the real mass changes and the resonant frequency is reliable. On the contrary, the characteristic ratio of a net density/viscosity effect is ∼10 Hz Ω^−1^, which is a characteristic of a net liquid-loading effect.

## 5. Digital Video Electrochemistry (DVEC)

During the last few years, a new strategy has been developed for the interpretation of electrochemical results associated with color changes. Digital Video Electrochemistry takes advantage of a fast CCD sensor of a Digital Video Camera that records video at 30 fps (or more). This technique has been developed both as a low-cost spectroelectrochemical device and a new technique allowing measuring the color intensity and color intensity dispersion on large electrode surfaces [89,90].

Video is decomposed into individual frames, and each frame is analyzed based on the RGB color theory. A digital image can be expressed as a matrix of pixels; each one of these pixels is characterized by its three-color coordinates in the RGB space. If a 3 × 8 bits camera is used, then a pure Red color corresponds to the (255,0,0) coordinates, a pure Green color (0,255,0), and a pure Blue color (0,0,255). As it is well known, the combination of all three coordinates gives 256 × 256 × 256 possible colors to be detected, which covers most of the visible spectra. An important difference with spectroelectrochemical techniques is that it is possible to simultaneously analyze a large surface area of the electrode. Since the entire surface can be simultaneously analyzed, there is a large amount of information to be managed, with three coordinates per pixel at any frame. Assuming a 30 fps 100 × 100 pixels image, that means 3 × 30 × 100 × 100 = 900,000 data per second. Therefore, the use of statistical parameters which summarize all this information should be necessary. For this purpose, and for the mean color intensities (${\bar{I}}_{RGB})$, Equation (39), and the standard deviation of color intensity for each one of the RGB coordinates (variance, $sd_{RGB}^{2}$), Equation (40) is a good option [89].
(39)$${\bar{I}}_{RGB}=\sum_{j=1}^{n_{pixel}}\frac{I_{RGB}\left(j\right)}{n_{pixels}}$$
(40)$$sd_{RGB}^{2}=\frac{\sum_{j=1}^{n_{pixels}}{\left(I_{RGB}\left(j\right)-{\bar{I}}_{RGB}\right)}^{2}\;}{n_{pixels}}$$

In recent works, it has been proven that these parameters can be directly related to kinetic parameters [89,90]. During a voltammetric experiment, this research has shown that the time derivative of color intensity showed similar peaks to voltammetric ones, and if there are several overlapped processes associated with different color changes, these peaks can help to separate the current contribution to each one of these processes (see Figure 6). In the case of electrocatalytic reactions, there is a current that does not cause color changes, and with the help of the time derivative of mean color intensity curves, it is possible to separate the electrocatalytic current from the current due to electrochemical processes of the surface-adsorbed species.

A second application for digital video analysis, together with the electrochemical response, is from the color dispersion analysis. Let us assume that during a surface chemical/electrochemical reaction, there is a color change from color 1 to color 2. Then, there is an initial situation where all the surface is color 1, and the final surface is color 2. Color dispersion proves minimum at the beginning and at the end of the chemical reaction, but larger values are observed at intermediate times [89,90]. Maximum of color dispersion is achieved if one-half of the surface is color 1 and the other half is color 2. That corresponds to the half-life time ($t_{1/2}$) for a kinetic study and is directly related to the kinetic constant for the studied process in such experimental conditions. Since the standard deviation of the color intensity on the electrode’s surface is the measurement of the color dispersion, the time at which this parameter proves maximum is an easy way to estimate the kinetic constant associated with the chemical/electrochemical process. This possibility has been used to study the evolution of the reduction kinetic constants of Prussian blue to Everitt’s salt from the variance maxima with time during the electrochemical process [90]. Moreover, the variance against time experimental curves show well-defined peaks, and from the half-peak width, it can be possible to discern the studied reaction among different kinetic orders [89].

More recently, digital video electrochemistry analysis has been used for the study of electroplating and electrodeposition processes. It is true that image analysis has been used for the characterization of electroplated surface [91], but the main advantage of DVEC is that this analysis takes place in situ and not after the surface has been electroplated. Therefore, it proves more adequate for the study of the kinetics and mechanism of the electrodeposition process. In this case, the maximum variance against time curves can be interpreted as the time needed to cover ½ of the electrode’s surface [92]. It is an excellent parameter to estimate the rate of the electrodeposition process. In electroplating studies, the electrodeposition on the electrode’s surface does not necessarily mean a color change. As an example, during the electrodeposition of copper on the surface of the electrode, if copper is deposited on a non-covered region of the electrode, the mean color intensity and the variance of color intensity change, but if copper is deposited on previously deposited copper, there is no color change, and probably, a non-homogeneous layer of copper on the electrode is formed [92]. Therefore, and if compared with electrodeposition followed by mass changes, with this technique, we can estimate the degree of the covered surface but not the total amount of electrodeposited material. It should be noted that the homogeneity of the electrode’s surface cannot be estimated from classical spectroelectrochemistry since this last technique usually gets the spectroscopic information on a small region on the electrode’s surface (or, in some cases, two regions of the electrode’s surface [93]). An important advantage of DVEC is that it is fully compatible with EQCM measurements, and then mass changes together with color dispersion and color intensity can be simultaneously measured and analyzed.

Another interesting parameter that can be obtained from DVEC is the time at which the surface is fully covered by the new layer [92]. For this purpose, there are two parameters that can be used. On the one side, there is the time from which the mean color intensity does not change. After this time, all the electrogenerated material is deposited on previously generated material, and then, no color change is measured. On the other side, there is the color dispersion analysis. If there are no new regions on the electrode covered by the electrogenerated material, then there is no change in the color dispersion, and the variance keeps constant. In both cases, this is the time needed to achieve a fully covered surface.

The study of electrodissolution by DVEC and electrochemical techniques of a previous Cu deposit has allowed identifying different Cu compounds formed on the electrode’s surface, for example, a white-grey compound corresponding to the formation of a CuCl layer during the electrodissolution of the last layer [92]. By comparing the time needed for the electrodeposition with the time needed for the electrodissolution, a rough estimation of the electrochemical efficiency was also obtained. The rate of covering the electrode’s surface has been measured at different parts on the electrode’s surface thanks to the local image analysis provided by the DVEC, and it has been proved that the ohmic drop of the electrode is a key factor determining the homogeneity of the covered surface [92].

More recently, the *ac*-response of DVEC has also been studied [29], obtaining the electrochromic performance of PEDOT films at different frequencies and at different potentials for the three-color coordinates, red, green, and blue.

## 6. Discussion and Conclusions

Electrochemical processes proved to be very complex and require the use of sophisticated techniques and methodologies to be interpreted correctly. The use of hyphenated techniques has proven to be one of the best strategies for this purpose. The simultaneous acquisition of several signals represents a qualitative and quantitative leap in the study of reaction mechanisms in electrochemical processes. When the measured signals can be related to changes in the concentration of the electroactive species, this implies the possibility of relating directly to the measured faradaic current. In this way, it is possible to identify species that are inserted by their molecular mass, reaction intermediates by their molar absorptivity, and catalytic processes that cause electric current but do not carry associated changes in mass or color. Acquiring the motional resistance during an electrochemical experiment has allowed to detect viscous changes at the electrode/solution interface. Although many of these coupled techniques do not have the same resolution as others, such as AFM, STM, SEM, or EDX, the possibility of obtaining a simultaneous response to the electrochemical response makes them ideal techniques for the kinetic characterization of electroactive materials or the identification of intermediate species in more complex mechanisms. It should be noted that in recent years, the use of video electrochemical techniques (DVEC) has introduced the possibility of statistically studying the changes that take place on the surface of an electrode. The variance versus time curves during a voltammetry experiment show a similar variation to the current curves and allow a new interpretation of the current measurement related to the dispersion of active centers on the surface.

## Figures and Tables

**Figure 1 materials-16-04226-f001:**
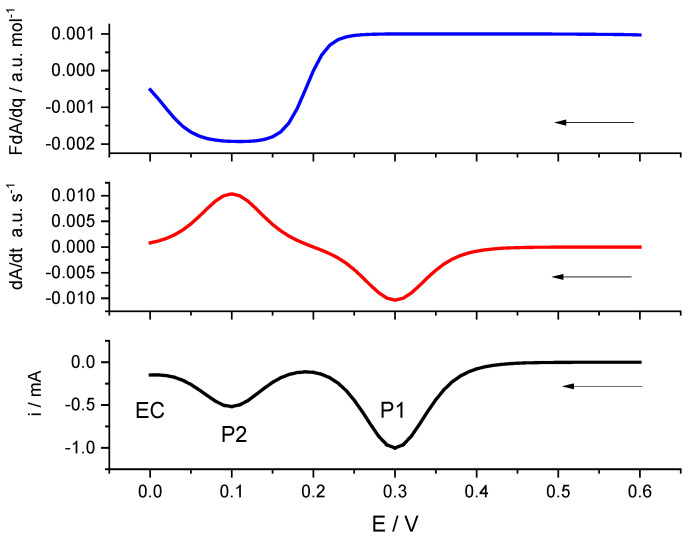
Simulated voltammetric curve for a reduction process of an absorbed specie, time derivative of absorbance and the electrochromic efficiency (FdA/dq) for two electrochromic centers, and an electrocatalytic process for the smaller potentials. EC refers to the electrocatalytic process, while P1 and P2 to the electrochemical processes associated with electroactive centers (1) and (2), respectively. Current was simulated from Equations (15) and (16) assuming two electrochemical processes with these parameters: $E_{1}^{0^{\prime }}=0.3\;V,$
$E_{1}^{0^{\prime }}=0.1\;V;$
$E_{s}=0.6\;V,$
$b=38.9{\;V}^{-1};$
$v=0.02\;{\mathrm{Vs}}^{-1};$
$n_{1}=n_{2}=1;$
$i_{m.1}=-0.001\;A;$
$i_{m,2}=-0.0005\;A.\;$ A simulated catalytic reaction was added as $i_{c}=-0.0001\exp\left(-\frac{0.5F}{RT}\left(E_{s}+vt-0.005\right)\right).\;dA/dt$ curves and FdA/dq curves were obtained, assuming k(λ) values for both individual processes as 0.001 a.u. and −0.002 a.u.

**Figure 2 materials-16-04226-f002:**
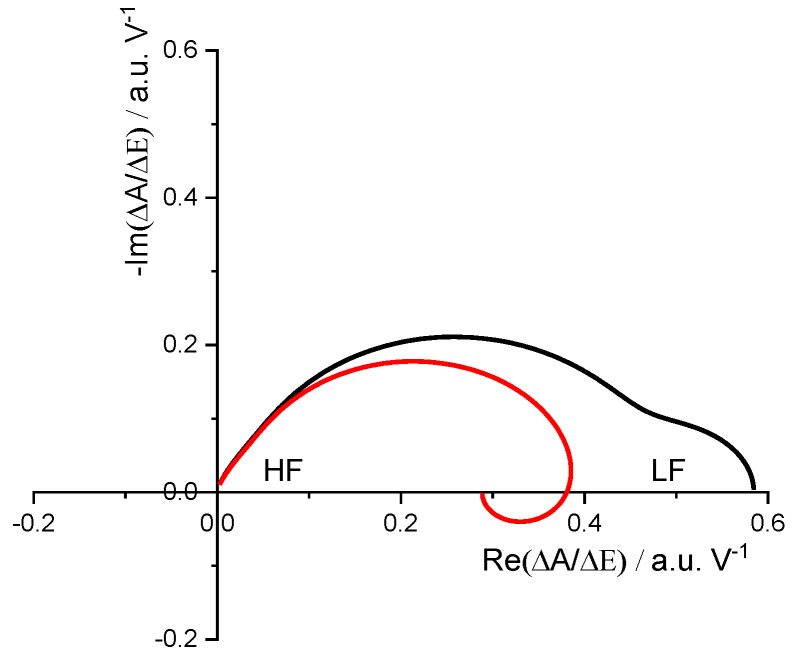
Simulated color impedance spectra for a system where 2 electroactive centers contribute to the color change at a given wavelength. Black Line: the faster process contributes positively, and the slower one negatively if the applied potential increases. Red Line: the faster and the slower process contribute positively. HF refers to high frequency, and LF to low frequency. Simulated curves were obtained from Equation (23) with these parameters: $\tau_{1}=0.08\;s,$
$\tau_{2}=1.6\;s;$
$k_{1}\left(\lambda \right)G_{1}=3.3\times 10^{-4}\;V^{-1}{\;s}^{-1}\;\mathrm{mol}\cdot L^{-1};$
$k_{2}\left(\lambda \right)G_{2}=2.2\times 10^{-5}\;V^{-1}{\;s}^{-1}\;\mathrm{mol}\cdot L^{-1},$
$K_{1}=6.0\;m\cdot s^{-1},$
$K_{2}=23\;m\cdot s^{-1}$. For the black line, the same parameters, except $k_{2}\left(\lambda \right)G_{2}=-2.2\times 10^{-5}\;V^{-1}{\;s}^{-1}\;\mathrm{mol}\cdot L^{-1}$.

**Figure 3 materials-16-04226-f003:**
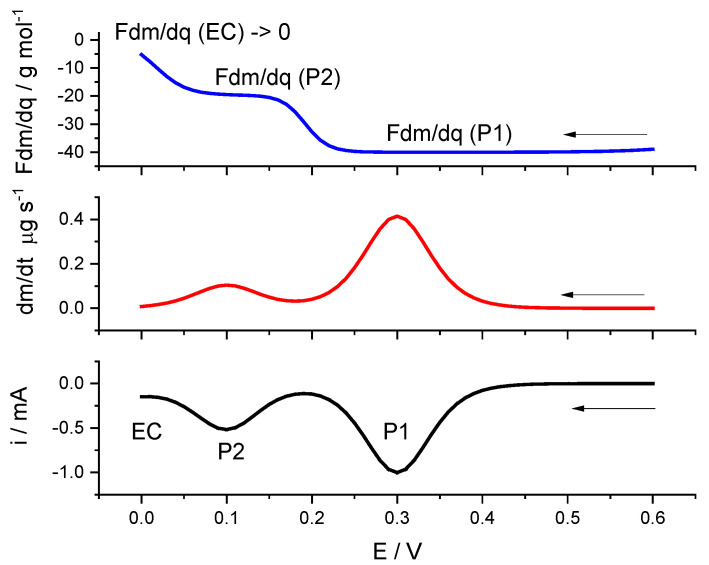
Simulated voltammetric curve for two electrochemical processes with different cation exchanges (P1, P2) and an electrocatalytic process (EC) at the smaller potentials. Current was simulated from Equations (15) and (16), assuming two electrochemical processes with these parameters: $E_{1}^{0^{\prime }}=0.3\;V,$
$E_{1}^{0^{\prime }}=0.1\;V;$
$E_{s}=0.6\;V;$
$b=38.9{\;V}^{-1};$
$v=0.02\;{\mathrm{Vs}}^{-1};$
$n_{1}=n_{2}=1,$
$i_{m.1}=-0.001\;A;$
$i_{m,2}=\;-0.0005\;A$. A simulated catalytic reaction was added as $i_{c}=-0.0001\exp\left(-\frac{0.5F}{RT}\left(E_{s}+vt-0.005\right)\right).\;dm/dt$ curves and Fdm/dq curves were obtained, assuming −$MW_{i}/z_{i}$ values for both individual processes as −$40\;g\cdot {\mathrm{mol}}^{-1}$ and −$20\;g\cdot {\mathrm{mol}}^{-1}$.

**Figure 4 materials-16-04226-f004:**
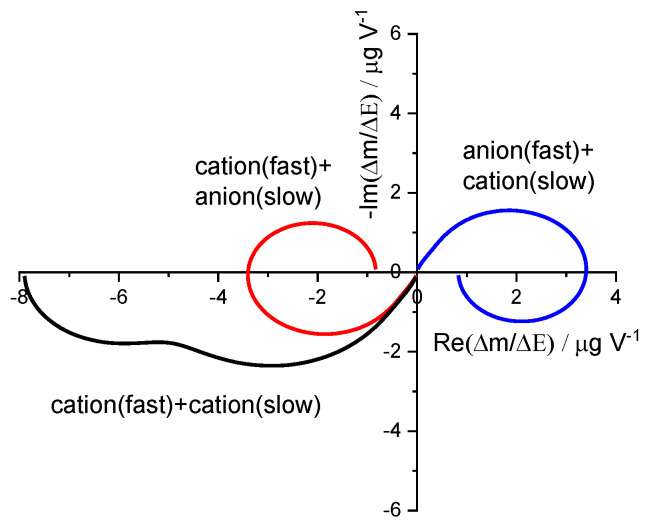
Simulation of *ac*-electrogravimetry for different cases: anion fast + cation slow (Blue line); cation + cation (Black line); and cation fast + anion slow (Red line). Simulated curves were obtained from Equation (35) with these parameters: $\tau_{1}=0.08\;s,$
$\tau_{2}=1.6\;s;$
$\frac{MW_{1}G_{1}}{z_{1}}=\pm 3.3\times 10^{-4}\;{g\;V}^{-1}{\;s}^{-1}\;\mathrm{mol}\cdot L^{-1};$
$\frac{MW_{2}G_{2}}{z_{2}}=\pm 5.2\times 10^{-5}{\;g\;V}^{-1}{\;s}^{-1}\;\mathrm{mol}\cdot L^{-1};$
$K_{1}=6.0\;m\cdot s^{-1},$
$K_{2}=23\;m\cdot s^{-1}1,$
$MW_{i}/z_{i}$ are positive for anions participation and negative for the cations participation.

**Figure 5 materials-16-04226-f005:**
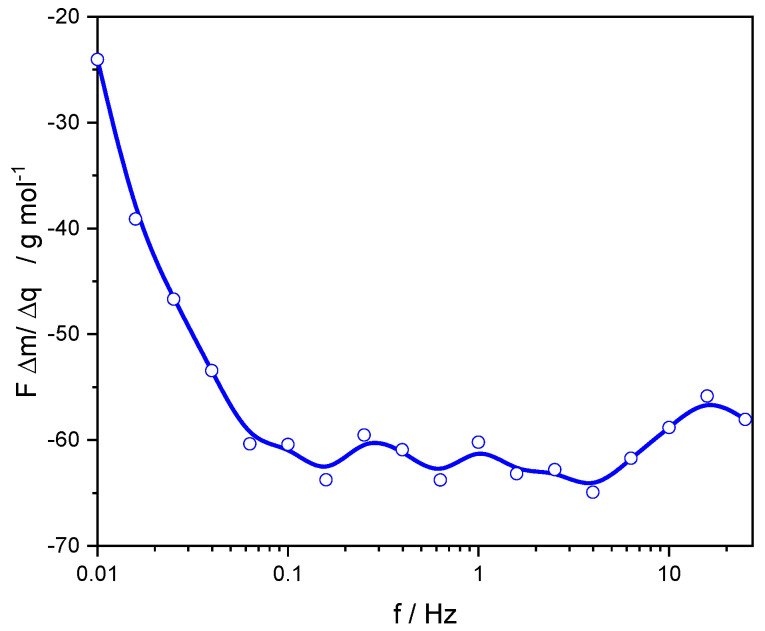
Dependence of FΔm/Δq on the frequency during am *ac*-electrogravimetric study of PEDOT films in 0.1 M ${\mathrm{LiClO}}_{4}\;$ aqueous solution [55].

**Figure 6 materials-16-04226-f006:**
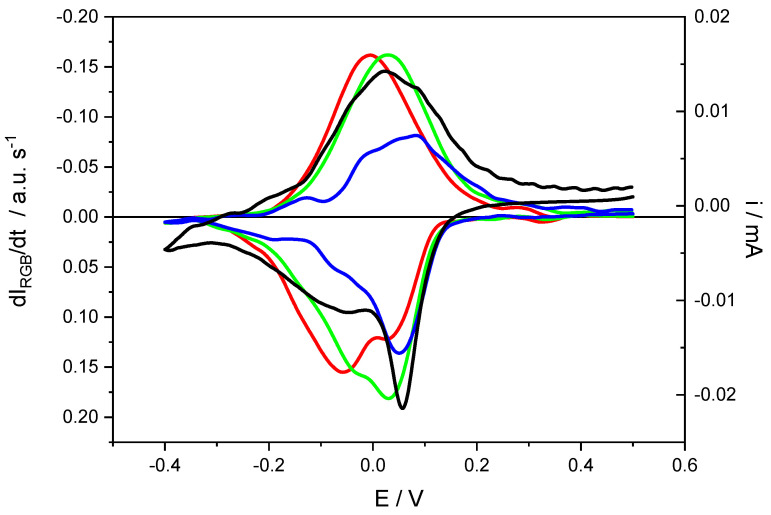
Cyclic voltammetry at 2 mV/s of a thin layer of Poly(methylene Blue) on an ITO transparent electrode in a KNO_3_ pH = 2.2 aqueous solution. Red, Green and Blue lines refer to Red, Green and Blue RGB coordinates, respectively.

## Data Availability

Available under demand.

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
