# Peer review of "Overview and Recent Advances in Hyphenated Electrochemical Techniques for the Characterization of Electroactive Materials"

_materials, 2023, doi:10.3390/ma16124226_

Round 1

Reviewer 1 Report

Dear authors

I have overall enjoyed article reading. The topic discussed by the authors is interesting to the audience, and in general, the article is well written. I list below some minor changes that must be addressed before further article processing.

Equations 7 through 12: The surface concentration symbol (Γ) is used recurrently in such equations. However, the subindex is not always defined, e.g. Γ???, Γ?? and so on. Please make sure that every symbol defined in the manuscript is defined before using it, or immediately after defining it. This observation also applies to the symbols from Equations 19 and 20.

Figure 1: The figure caption claims the voltametric curve was obtained on the basis of a simulation; what were the simulation parameters that yielded the curve? This observation also applies to the simulation plots of figures 2, 3 and 4.

Section “Corrosion and formation of passive layers”. There are unnumbered equations in this section.

Author Response

First of all, we would like to thank the reviewers of this manuscript for their work and time. We accept the proposed criticisms and suggestions as they are constructive criticisms and the proposed changes will undoubtedly improve the quality of the manuscript.

Referee 1

Dear authors

I have overall enjoyed article reading. The topic discussed by the authors is interesting to the audience, and in general, the article is well written. I list below some minor changes that must be addressed before further article processing.

Equations 7 through 12: The surface concentration symbol (Γ) is used recurrently in such equations. However, the subindex is not always defined, e.g. Γ???, Γ?? and so on. Please make sure that every symbol defined in the manuscript is defined before using it, or immediately after defining it. This observation also applies to the symbols from Equations 19 and 20.

Thank you very much for this observation. We have corrected this defect in all the equations raised, as well as in the new equations of the revised version.

Figure 1: The figure caption claims the voltametric curve was obtained on the basis of a simulation; what were the simulation parameters that yielded the curve? This observation also applies to the simulation plots of figures 2, 3 and 4.

We have changed the figure captions for all simulations indicating the parameters used in the simulations. We have also introduced in the text the equations used for the simulation of the different curves. In the case of the color impedance and mass impedance simulations we have changed the figures to mark numerical scales on the axes in order to a better understanding of these simulated curves.

Section “Corrosion and formation of passive layers”. There are unnumbered equations in this section.

We have corrected it.

Reviewer 2 Report

The manuscript is summarizing previous literature on hyphenated electrochemical techniques. The review is well-referenced and reports interesting data related to an important issue (e.g., characterization of electroactive materials), but needs some changes. For example, some case studies discussion can enhance the quality of the review and make it more attractive. You have about 100 references, take the latest studies on different topics and analyze them and report how these techniques were used, and integrate them with your review

Author Response

First of all, we would like to thank the reviewers of this manuscript for their work and time. We accept the proposed criticisms and suggestions as they are constructive criticisms and the proposed changes will undoubtedly improve the quality of the manuscript

Referee 2

The manuscript is summarizing previous literature on hyphenated electrochemical techniques. The review is well-referenced and reports interesting data related to an important issue (e.g., characterization of electroactive materials), but needs some changes. For example, some case studies discussion can enhance the quality of the review and make it more attractive. You have about 100 references, take the latest studies on different topics and analyze them and report how these techniques were used, and integrate them with your review.

Thank you for this note. We also think that presenting some real case studies improves the quality of the manuscript. In the new version of the manuscript we have added text explaining how we apply the different analysis strategies to characterize these electroactive materials and their electrochemical properties. For example, in the analysis of absorbance derivative curves with time during voltammetric experiments we have explained how we were able to separate the contribution of different active centers to the overall electrochemical response for the Poly(azure A)[1]. In another example, we explain how this analysis allows us to separate the contribution of the catalytic current of dissolved oxygen reduction during the reduction of PEDOT films[2].

Another case analyzed is that of color impedance to detect different active centers by their relative velocity. The presence of more than one semicircle in the color impedance diagrams has allowed the discovery of different active sites for the insertion of ions in this material [3].

We also analyze the case of an anodic dissolution mechanism of bivalent metals and how by means of the Fdm/dq function it is possible to distinguish whether the dissolution of the metal takes place after the first or the second electron transfer [4–6].

A mass impedance study of PEDOT films in LiClO4 has also been included. From the use of the transfer function  it is possible to separate the fast contribution associated with the electrical charge and discharge of the polymer from the low frequency response, possibly associated with catalytic oxygen reduction processes [2].

Finally, two cases of application of the DVEC technique coupled with electrochemical techniques have been introduced. The first one is a voltammogram of a Poly(methylene blue) film together with the changes of the color intensity derivative for the three coordinates, Red, Green and Blue. These are unpublished results and are an example of how the color derivative peaks coincide with different peaks in the current response of this system. The second case corresponds to the study of Cu electrodeposition on a resistive composite material. In this case, the coupling of DVEC with electrochemical techniques has allowed us to characterize the growth rate of the Cu layer on the surface, as well as the formation of some salt during dissolution. Surface color changes are an excellent indicator to detect the formation of compounds and intermediates[7].

References

  1. Agrisuelas, J.; Giménez-Romero, D.; García-Jareño, J.J.; Vicente, F. Vis/NIR Spectroelectrochemical Analysis of Poly-(Azure A) on ITO Electrode. Electrochem. Commun. 2006, 8, 549–553, doi:10.1016/j.elecom.2006.01.022.
  2. Guillén, E.; Agrisuelas, J.; García-Jareño, J.J.; Vicente, F. The Role of Lithium, Perchlorate and Water during Electrochemical Processes in Poly(3,4-Ethylenedioxythiophene) Films in LiClO4 Aqueous Solutions. J. Electroanal. Chem. 2021, 897, 115580, doi:10.1016/j.jelechem.2021.115580.
  3. Agrisuelas, J.; García-Jareño, J.J.; Vicente, F. Identification of Processes Associated with Different Iron Sites in the Prussian Blue Structure by in Situ Electrochemical, Gravimetric, and Spectroscopic Techniques in the Dc and Ac Regimes. J. Phys. Chem. C 2012, 116, 1935–1947, doi:10.1021/jp207269c.
  4. Gimenez-Romero, D.; Garcia-Jareno, J.J.; Vicente, F. EQCM and EIS Studies of Zn-Aq(2+)+2e(-) Reversible Arrow Zn0 Electrochemical Aq Reaction in Moderated Acid Medium. J. Electroanal. Chem. 2003, 558, 25–33, doi:10.1016/S0022-0728(03)00373-5.
  5. Gimenez-Romero, D.; Gabrielli, C.; Garcia-Jareno, J.J.; Perrot, H.; Vicente, F. Electrochemical Quartz Crystal Microbalance Study of Copper Electrochemical Reaction in Acid Medium Containing Chlorides. J Electrochem Soc 2006, 153, J32–J39.
  6. Gregori, J.; Garcia-Jareno, J.; Gimenez-Romero, D.; Roig, A.; Vicente, F. Anodic Dissolution of Nickel across Two Consecutive Electron Transfers - Calculation of the Ni(I) Intermediate Concentration. J. Electrochem. Soc. 2007, 154, C371–C377, doi:10.1149/1.2737665.
  7. Agrisuelas, J.; García-Jareño, J.J.; Guillén, E.; Vicente, F. RGB Video Electrochemistry of Copper Electrodeposition/Electrodissolution in Acid Media on a Ternary Graphite:Copper:Polypropylene Composite Electrode. Electrochimica Acta 2019, 305, 72–80, doi:10.1016/j.electacta.2019.03.016.

Reviewer 3 Report

 Reviewer comments

This article discusses the evolution of the hyphenated electrochemical techniques to extract important information for electroactive material characterisation. Electrochemical process kinetics have been studied using this method by the ac-regime. Understanding of electrode processes has been improved by estimating the molar masses of ex-changed species or apparent molar absorptivities at different wavelengths.

There are numerous typos and grammatical errors. The literature is not cohesive, although it provides significant emphasis on hyphenated electrochemical techniques for electroactive material characterisation. The manuscript could be considered for publication in "Materials" with the modifications recommended in the comments. Here are my comments.

Comments:

1.     To expedite the review process, authors or journal should include line numbers.

2.     Contrary to the title of the manuscript, more emphasis is laid on hyphenated electrochemical techniques than on materials analysed using these techniques. There must be a correlation between the techniques and the electroactive material being analysed.

3.     Explain the abbreviated terms  such as I, dq, dn, dt,  ? ??/?? or ? ??/?? etc.  throughout the manuscript for better comprehension of the readers.

4.     "Recording data and before data processing" should be a subheading (1.1) or a part of the paragraph. Revise all such entries.

5.     Provide citations for equations used throughout the manuscript.

6.     Provide the citation for figures reproduced or replotted from another journal at the end of the caption.

7.     When paired with other (spectroscopic) approaches, the majority of the techniques discussed here seem to be advantageous for measuring electrochemical processes. However, the manuscript fails to discuss the drawbacks of these coupling strategies. For instance, in the study of Cu electrodissolution by DVEC, colour change for Cu is referred as one of the measurement criteria. However, despite the obvious physical change, there is no need to assess this property, as colour change may be caused by a variety of factors, obscuring its significance as information. Throughout the text, there are other similar concerns. Describe each of these in the revised version.

8.     Eliminate the use of redundant words e.g., where, however, thus, in general etc. Revise all similar cases, as removing these term(s) would not significantly affect the meaning of the sentence.

9.     There are numerous self-citations by authors, adjust them as per “Material” predefined policy on self-citations.

Need moderate editing

Author Response

First of all, we would like to thank the reviewers of this manuscript for their work and time. We accept the proposed criticisms and suggestions as they are constructive criticisms and the proposed changes will undoubtedly improve the quality of the manuscript.

Referee 3

This article discusses the evolution of the hyphenated electrochemical techniques to extract important information for electroactive material characterisation. Electrochemical process kinetics have been studied using this method by the ac-regime. Understanding of electrode processes has been improved by estimating the molar masses of ex-changed species or apparent molar absorptivities at different wavelengths.

There are numerous typos and grammatical errors. The literature is not cohesive, although it provides significant emphasis on hyphenated electrochemical techniques for electroactive material characterisation. The manuscript could be considered for publication in "Materials" with the modifications recommended in the comments. Here are my comments.

Comments:

  1. To expedite the review process, authors or journal should include line numbers.

We use a template from the publisher for manuscript preparation, we do not know how to add the line number. We are sorry, but we do not know how to add the line number.

  1. Contrary to the title of the manuscript, more emphasis is laid on hyphenated electrochemical techniques than on materials analysed using these techniques. There must be a correlation between the techniques and the electroactive material being analysed.

In the revised version of the manuscript we have modified part of the discussion and added information on the properties of the materials that have been characterized in the cited studies. Since the work is a review, it is difficult to go in depth into the characterization of the materials presented in all the examples. To see in more detail the properties of the materials, it is better to refer to each of the bibliographic citations. Let us not forget either that since these techniques are coupled to electrochemical techniques, what is really characterized is the electrochemical behavior of these materials during the electrochemical processes.

  1. Explain the abbreviated terms  such as I, dq, dn, dt,  ???/?? or ? ??/??  throughout the manuscript for better comprehension of the readers.

Thank you for this note, we have tried to identify all symbols and functions during their appearance in the text in the new version of the manuscript.

  1. "Recording data and before data processing" should be a subheading (1.1) or a part of the paragraph. Revise all such entries.

Thanks again, we have corrected

  1. Provide citations for equations used throughout the manuscript.

Thanks, we have added citations for equations in the text.

  1. Provide the citation for figures reproduced or replotted from another journal at the end of the caption.

Thanks, there are no figures reproduced from another journal. Only Figure 5 corresponds to a parallel study of PEDOT films in similar but different experimental conditions of reference [2]

  1. When paired with other (spectroscopic) approaches, the majority of the techniques discussed here seem to be advantageous for measuring electrochemical processes. However, the manuscript fails to discuss the drawbacks of these coupling strategies. For instance, in the study of Cu electrodissolution by DVEC, colour change for Cu is referred as one of the measurement criteria. However, despite the obvious physical change, there is no need to assess this property, as colour change may be caused by a variety of factors, obscuring its significance as information. Throughout the text, there are other similar concerns. Describe each of these in the revised version.

We have added in the text a small discussion about the real usefulness of each coupling. In the case of DVEC we have already considered that it is only useful to see color changes during the deposition of the first layers, after that there will be no color change. However, we do not agree that color changes associated with other processes can be a serious problem for the applicability of the technique, on the contrary, as the color is characterized by its three RGB coordinates, these changes can be used to detect other species or intermediates during the process, as it has happened during the dissolution of Cu in HCl[7]. Regarding the use of the Fdm/dq or FdA/dq functions we do have a problem in the regions where the current is close to 0 since an asymptotic behavior appears that prevents a correct analysis. This problem has been described in the revised version of the manuscript.

  1. Eliminate the use of redundant words e.g., where, however, thus, in general etc. Revise all similar cases, as removing these term(s) would not significantly affect the meaning of the sentence.

We have revised the English of the manuscript in an attempt to simplify many of these expressions.

  1. There are numerous self-citations by authors, adjust them as per “Material” predefined policy on self-citations.

This is true and we have cut out most of the self-citations from the manuscript. We have also added bibliography from other authors with similar cases to the bibliography we have removed.

References

  1. Agrisuelas, J.; Giménez-Romero, D.; García-Jareño, J.J.; Vicente, F. Vis/NIR Spectroelectrochemical Analysis of Poly-(Azure A) on ITO Electrode. Electrochem. Commun. 2006, 8, 549–553, doi:10.1016/j.elecom.2006.01.022.
  2. Guillén, E.; Agrisuelas, J.; García-Jareño, J.J.; Vicente, F. The Role of Lithium, Perchlorate and Water during Electrochemical Processes in Poly(3,4-Ethylenedioxythiophene) Films in LiClO4 Aqueous Solutions. J. Electroanal. Chem. 2021, 897, 115580, doi:10.1016/j.jelechem.2021.115580.
  3. Agrisuelas, J.; García-Jareño, J.J.; Vicente, F. Identification of Processes Associated with Different Iron Sites in the Prussian Blue Structure by in Situ Electrochemical, Gravimetric, and Spectroscopic Techniques in the Dc and Ac Regimes. J. Phys. Chem. C 2012, 116, 1935–1947, doi:10.1021/jp207269c.
  4. Gimenez-Romero, D.; Garcia-Jareno, J.J.; Vicente, F. EQCM and EIS Studies of Zn-Aq(2+)+2e(-) Reversible Arrow Zn0 Electrochemical Aq Reaction in Moderated Acid Medium. J. Electroanal. Chem. 2003, 558, 25–33, doi:10.1016/S0022-0728(03)00373-5.
  5. Gimenez-Romero, D.; Gabrielli, C.; Garcia-Jareno, J.J.; Perrot, H.; Vicente, F. Electrochemical Quartz Crystal Microbalance Study of Copper Electrochemical Reaction in Acid Medium Containing Chlorides. J Electrochem Soc 2006, 153, J32–J39.
  6. Gregori, J.; Garcia-Jareno, J.; Gimenez-Romero, D.; Roig, A.; Vicente, F. Anodic Dissolution of Nickel across Two Consecutive Electron Transfers - Calculation of the Ni(I) Intermediate Concentration. J. Electrochem. Soc. 2007, 154, C371–C377, doi:10.1149/1.2737665.
  7. Agrisuelas, J.; García-Jareño, J.J.; Guillén, E.; Vicente, F. RGB Video Electrochemistry of Copper Electrodeposition/Electrodissolution in Acid Media on a Ternary Graphite:Copper:Polypropylene Composite Electrode. Electrochimica Acta 2019, 305, 72–80, doi:10.1016/j.electacta.2019.03.016.

Round 2

Reviewer 2 Report

The authors answered my questions satisfactorily. The manuscript can be published in its current form.

Reviewer 3 Report

The amendments made by the authors in response to the reviewers' comments are adequate, hence the paper is accepted for publication.